# ContraMTD: An Unsupervised Malicious Network Traffic Detection Method based on Contrastive Learning

## ABSTRACT

Malicious traffic detection has been a focal point in the field of network security, and deep learning-based approaches are emerging as a new paradigm. However, most of them are supervised methods, which highly depend on well-labeled data, and fail to handle unknown or continuously evolving attacks. Unsupervised methods alleviate the need for labeled data, but existing methods are often limited to detecting anomalies either in vertical perspective through historical comparisons or in horizontal perspective by comparing with concurrent entities. Relying on data from a single perspective is unreliable, and it limits the model's accuracy and generalizability. In this paper, we propose a novel method ContraMTD based on contrastive learning, which comprehensively considers both vertical and horizontal perspectives. ContraMTD extracts local behavior features and global interaction features from normal network traffic by proposed SEC and DE-GAT respectively, then employs contrastive learning to learn the relationship, especially consistency between them, and finally detects malicious traffic through a multi-round scoring approach. We conduct extensive experiments on three datasets, including a self-collected dataset, and the results demonstrate that our method outperforms many state-of-the-art methods in the domain of unsupervised malicious traffic detection.

## CCS CONCEPTS

• **Security and privacy** → **Network security**; • **Computing methodologies** → **Artificial intelligence**; • **Information systems** → **Data mining**.

## KEYWORDS

Malicious network traffic detection, Contrastive learning, Graph neural network

**ACM Reference Format:**
. 2018. ContraMTD: An Unsupervised Malicious Network Traffic Detection Method based on Contrastive Learning. In *Proceedings of Make sure to enter the correct conference title from your rights confirmation emai (Conference acronym 'XX)*. ACM, New York, NY, USA, 9 pages. https://doi.org/XXXXXXX.XXXXXXX

## 1 INTRODUCTION

Over the past few decades, malicious network traffic, encompassing Distributed Denial-of-Service (DDoS) attacks, network scanning, and data exfiltration, has posed a severe threat to the cybersecurity of individuals, enterprises, and even nations. Accurate detection of malicious traffic always receives widespread attention from both academic and industrial communities. Traditional rule based methods [19, 22, 29] and deep packet inspection (DPI) based methods [17, 21] have become ineffective due to the continually evolving attacks and the wide use of encryption techniques.

In order to detect the increasingly sophisticated attacks, machine learning (like Support Vector Machine, Decision Tree) based methods [12, 16, 23] and deep learning (like Convolutional Neural Network, Transformer, Graph Neural Network) based methods [9, 11, 14] have been proposed. The detection accuracy of some of them can be as high as 99%. They can automatically learn the latent features from high-dimensional data on large-scale datasets, thereby enhancing their detection performance, especially for those based on deep learning. However, most of them are supervised, and their remarkable performance highly depends on well-labeled datasets, which will encounter many challenges in real-world scenarios. (1) *High cost of high-quality dataset labeling.* A qualified network traffic dataset requires millions of packets for reliability [28], and the process of cleaning and labeling is laborious, time-consuming, and demands specialized expertise. (2) *Continuing evolution of attacker behaviors.* Attackers continually update and refine their techniques, introducing attack variants and even 0-day exploits [4]. Supervised methods are unable to detect these previously unseen malicious traffic patterns.

Unsupervised malicious network traffic detection approaches address the above challenges to some extent. These methods usually focus on either *vertical* analysis, where an entity's behavior is evaluated against historical data to identify anomalies that do not conform to normal patterns, or *horizontal* analysis, wherein an entity's behavior is compared with that of other entities in the same time frame and environment to pinpoint anomalies that deviate significantly from the majority. However, relying on data from a single perspective is unreliable. Specifically, normal network traffic evolves over time, rendering past data unrepresentative of the current situation. Additionally, concurrent data can be unstable due to various influencing factors and may be constrained by a limited number of samples involved. This leads to limited generalization ability and accuracy of the model. Therefore, a detection method that comprehensively considers both horizontal and vertical perspectives is needed.

We introduce contrastive learning [6] to solve the above problem. Contrastive learning is a powerful self-supervised technique that thrives on comparing different samples to learn useful feature representations and the matching of pairs without the need of manual labels. To the best of our knowledge, this is the first time contrastive learning has been applied to unsupervised malicious traffic detection. We observe that normal behaviors in a network not only exhibit a certain level of stability themselves but also show inherent correlations with other behaviors they interact with. All

of these are closely tied to specific objectives or tasks, leading to the consistency between local behavior features and global interaction features. However, attack behaviors often break the consistency. Based on this, we propose a novel method called ContraMTD. We elaborately construct sample pairs utilizing local and global features, employing contrastive learning to learn the consistency from historical data, thereby enabling us to effectively identify anomalies that deviate from pre-learned patterns in the vertical perspective. During the detection phase, we measure the similarity between features in both positive and negative sample pairs in multiple round, achieving detecting in the horizontal perspective.

Specifically, we divide network traffic into channels, and construct positive sample pairs using local behavior features and global interaction features that belong to the same channel, while construct negative sample pairs using features from different channels. For the local behavior feature, we introduce a SEC strategy, coupled with CNN, to capture robust and fine-grained features, including the sequence and density of behaviors. For the global interaction feature, we propose a Graph Double Edge Attention Network (DE-GAT) to extract features from the topology and attributes of host interaction multigraph. Besides, we mitigate the issue of false negative sample pairs during the contrastive learning process through K-means clustering. For the detection phase, we also introduce a multi-round scoring approach to enhance the stability of the results.

In conclusion, the main contributions of this paper are as follows:

- We apply contrastive learning to unsupervised malicious traffic detection for the first time, and leverage the consistency between local behavior features and global interaction features of network traffic for model training, which suggests a potential avenue for future exploration.
- We propose ContraMTD, which learns the consistency between two types of features of normal traffic as well as the relationships among normal traffic, and employs a multi-round scoring approach for anomaly detection, enabling a comprehensive approach for identifying malicious traffic across both vertical and horizontal perspectives.
- We introduce the SEC strategy, coupled with CNN, to learn fine-grained and robust local behavior features, and propose DE-GAT to learn global interaction features. We also mitigate the issue of false negative samples in contrastive learning by incorporating clustering techniques.
- We validate the effectiveness of ContraMTD through extensive experiments on three datasets with different scales and application scenarios, including a self-collected dataset, and the results demonstrate that our ContraMTD outperforms other methods.

## 2 METHODOLOGY

### 2.1 Overall Framework

Figure 1 illustrates the overall framework of ContraMTD, and it consists of five modules. Network Traffic Aggregation module first divides traffic into channels as processing units. Local Behavior Feature Extraction module and Global Interaction Feature Extraction module are employed to learn local features and global features of channels, respectively. Following feature extraction, Contrastive Learning module constructs positive and negative sample pairs,

learns the consistency from them and train the model. Finally, Anomaly Detection module constructs sample pairs for each sample to be tested and determine whether it is malicious through multi-round scoring.

### 2.2 Network Traffic Aggregation

Network traffic is a mixture of packets generated by hosts within a network. It is usually processed on the basis of a single packet[25] or divided into flows, sessions[14], or channels[7]. We partition network traffic into channels for processing in this paper. Packets with the same 5-tuples (source IP, destination IP, source port, destination port, protocol) constitute a flow, and a session $S = [P_{t_1}, P_{t_2}, ..., P_{t_{n_s}}]$ is composed of bidirectional flows, $P_{t_i}$ is the packet at time $t_i$, and $n_s$ is the number of packets in a session. Furthermore, The sessions between two hosts form a channel $C = [S_1, S_2, ..., S_{n_c}]$, and $n_c$ is the number of sessions in a channel.

A channel represents the interactions between two hosts over a period of time, which contains more comprehensive behavior information than flow. We thoroughly investigate its characteristics: (1) *Content.* The fundamental difference between channels is the content they transmit. It influences attributes such as packet length and the number of packets in a session. (2) *Temporal pattern.* The temporal pattern includes the sequence of sessions in a channel and their distribution across the time dimension. For example, individual web browsing is steady over time, while flood attack traffic is bursty and short-lived. (3) *Coherence.* The session in a channel inherently connects with its context, driven by a shared objective. For instance, a user's activities like logging in, uploading files, and database access on a server are coherent due to common objectives.

### 2.3 Local Behavior Feature Extraction

*2.3.1 Channel feature extraction.* To better explore the above characteristics of channels, we divide channels into segments, extract features from each segment, and then compress these segment features to get channel features. We refer to this process as SEC (segmentation, extraction, and compression) for convenience.

**Time-slot based channel segmentation.** We extend the concept of channel to *segment* to refine the feature of channels. A segment includes parts of sessions within a channel, and a channel can also be composed of several segments. Let the segment as $SG$, $t_s$ and $t_e$ are the start time and end time of $SG$ respectively, $SG$ is defined as:

$$SG = [S_1, S_2, ..., S_{n_{sg}}], t_{n_s}^1 < t_s \text{ and } t_1^{n_{sg}} < t_e \quad (1)$$

where $t_{n_s}^1$ is the timestamp of last packet in $S_1$, $t_1^{n_{sg}}$ is the timestamp of the first packet in $S_{n_{sg}}$, and $n_{sg}$ is the number of sessions in $SG$. Note that a session can belong to multiple segments in a channel, and the size of $SG$ can be 0.

We split the duration of a channel $T$ into $N_{sl}$ time slots of equal size $t_{sl} = T/N_{sl}$, and then group the sessions into the corresponding time slots to form segments. Therefore, a channel can also be represented as:

$$C = [SG_1, SG_2, ..., SG_{n_{cs}}] \quad (2)$$

where $n_{cs} = N_{sl}$ is the number of segments in a channel.

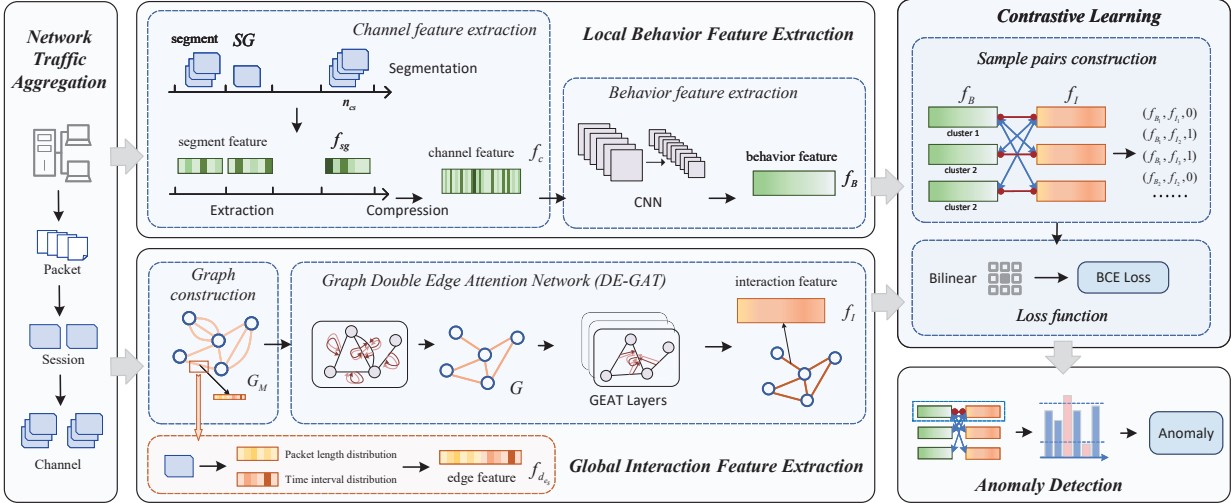

**Figure 1: Overview of ContraMTD Framework**

By segmenting channels into sessions, we can not only capture the inherent characteristics of individual sessions, but also gain insights into their temporal distribution such as the sequence of sessions and varying density of session occurrences.

**Segment feature extraction.** For each segment, we first extract the statistical features from each session, including the duration, and the count of forward and backward packets, and then derive segment features based on these statistical features. The segment feature $f_{sg}$ is a one-dimensional vector, and it contains the averages of forward and backward packet counts, the average of forward and backward packet byte sizes, the average and maximum session duration, and the average time intervals between packets. If the size of a segment is zero, all values within $f_{sg}$ will be set to 0.

**Segment feature compression.** In each channel, the features of all sessions together define the channel's features. The most straightforward method is to concatenate all segment features into a vector as behavior feature, but this approach has several problems. First, the new features are sparse because the interaction between hosts does not occur continuously, potentially increasing computational complexity. Second, the new features are not robust and are highly sensitive to time. Factors such as network latency or the randomness in the selection of channel start times can shift the positions of sessions within a channel, then the same behaviors might be represented by different features, even if the relative positions of sessions remain unchanged.

We address these issues by aligning and compressing segment features. Segment features that are entirely zeros are first removed. For the remaining, we retain the first $N_{cp}$ segment features and concatenate them into a sequence, noted as $f_{sq}$. If the length is less than $N_{cp}$, a zero vector is used for padding. The meta feature $f_{mt}$ of these segment features is appended to the end of the sequence to form the final channel feature $f_c = [f_{sq}, f_{mt}]$. $f_{mt}$ includes source port count, destination port count, and the number of segments with non-zero values.

*2.3.2 Behavior feature extraction.* To consider the correlation of features in $f_c$ and obtain more representative local behavior feature $f_B$, we reshape $f_c$ into a two-dimensional matrix as the input for a Convolutional Neural Network (CNN). CNN can capture the relationships between different elements at various levels of abstraction in a matrix through multiple kernels. The process is formalized as:

$$f_B = F_{CNN}(F_R(f_c)) \tag{3}$$

where $F_{CNN}$ represents the CNN, containing two convolutional neural layers, two pooling layers, and three fully connected layers. $F_R$ is the reshape operation.

## 2.4 Global Interaction Feature Extraction

*2.4.1 Graph construction.* In addition to its attribute information, each channel also has significant global semantic information through its interrelationships with other hosts or channels.

To capture the global interaction feature, we first construct a host interactive graph based on network traffic. We handle each session separately here to gain more complete information. $G_M = \{V, E_M, A_M\}$ is the host interaction graph, and it is a multigraph. $V$ is the vertex set, each node represents a host, identified by its IP address. $E_M = \{e_m | e_m = \{e_s | e_s = (u, v)\}, u \in V, v \in V\}$ is the edge set. $e_m$ denotes the multiple parallel edges between node $u$ and node $v$, and is a set of single edge $e_s$. Multiple parallel edges between a single pair of nodes correspond to multiple sessions from the same channel. $A_M = \{f_{de_m} | e_m \in E_M\}$ denotes the feature set of edges, in which $f_{de_m} = \{f_{de_s} | e_s \in e_m\}$ is the set of features of the simple edge $e_s$ between the host pair, and $f_{de_s}$ is the feature of each simple edge as described below.

For each edge, we extract the packet length distribution and time interval distribution between packets of the corresponding session and concatenate them as the edge feature $f_{de_s}$. The packet length reflects many information such as the content of network traffic, and the time interval involves the temporal characteristics of sessions. The distribution of these two features provides a more fine-grained

representation. Besides, to prevent the introduction of noise or information leakage for the contrastive learning process, the edge feature and channel feature need to be different. Specifically, we bin packet lengths into uniform intervals with a bucket size of 50, and divide time intervals into bins with logarithmically increasing sizes based on a base of 10.

*2.4.2 Global feature extraction.* To capture the unique characteristics of network traffic and host interaction graphs, and address the limitations of existing methods that are primarily tailored for simple graphs, we introduce a novel graph neural network, termed the Graph Double Edge Attention Network (DE-GAT), for extracting richer global interaction features.

**DE-GAT.** We first learn the interdependencies within sessions in each channel before capturing the interaction features between channels. To precisely capture these dependencies and make the model more focused on the sessions that play a key role in the channel, we employ the self-attention mechanism [30]. Through this mechanism, we calculate attention scores for the parallel edges between each pair of nodes and aggregate them using a weighted sum. Consequently, we merge these multiple parallel edges into a new edge, transforming the multigraph into a simple graph. The new edge representation is calculated as:

$$f_e = \frac{1}{n_c} \sum_{o=1}^{n_c} Att_o \tag{4}$$

$$Att = softmax(\frac{QK^T}{\sqrt{d_k}})V \tag{5}$$

where $Att$ is the representation of every single edge in the multigraph after the self-attention mechanism. The Query, Key, and Value are calculated as $Q = f_{d_{e_m}} W^Q$, $K = f_{d_{e_m}} W^K$ and $V = f_{d_{e_m}} W^V$, and $W^Q$, $W^K$, and $W^V$ are learnable weights, and $d_k$ is the dimension of single edge feature $f_{d_{e_s}}$, $n_c$ is the number of edges between a pair of nodes and the number of sessions in a channel.

After that, the multigraph $G_M$ is converted into a simple graph $G = \{V, E, A\}$, and $E = \{e | e = (u, v), u \in V, v \in V\}$ is the edge set, and $A = \{f_e | e \in E\}$ is the feature set of new edges.

Then, DE-GAT learns the global features for each edge by stacking multiple our newly proposed graph edge attention (GEAT) layers. The output of the final GEAT layer is the global interaction features $f_I$ of the channel.

**GEAT layer.** The global interaction feature $f_I$ is jointly determined by channel attributes, graph typology, and interdependent relationships between channels. To capture these information and focus on edge features, we elaborately design the GEAT layer inspired by graph attention neural network [31].

For each edge in graph $G$, GEAT layer assigns different weights to its neighboring edges and aggregates information from these neighbors. We also use the multi-head attention mechanism to learn multi-scale features. The process can be formally described as:

$$f_e^{(m+1)} = \|_{k=1}^{K} \sigma(\sum_{q \in \mathcal{N}(e)} \alpha_{e,q}^k W^k f_q^{(m)}) \tag{6}$$

$$\alpha_{e,q} = \frac{exp(\phi(W^e e_e \| W^e e_q))}{\sum_{p \in \mathcal{N}(e)} exp(\phi(W^e e_e \| W^e e_p))} \tag{7}$$

where $d_e^{(m+1)}$ is the edge feature output by the $m^{th}$ GEAT layer, and $d_e^{(0)}$ equals to $f_e$. $\mathcal{N}(\cdot)$ denotes the neighboring edges of an edge, and two edges are considered neighbors if they share a common node. $K$ is the number of heads in the multi-head attention mechanism. $\alpha_{e,q}^k$ is the attention value computed by the $k^{th}$ head. $\phi(\cdot)$ is the activation function, which is Leaky ReLU here. $\|$ represents the concatenation operation. $W^k$ and $W^e$ are learnable weights.

## 2.5 Contrastive Learning

For each normal channel in the network, its local behavior feature and global interaction feature are consistent because of the common motivation. We use contrastive learning technique to capture these pattern. In contrastive learning, positive and negative pairs are firstly formed, with positive pairs being similar and negative pairs being dissimilar, then the model is optimized using a contrastive loss function to minimize intra-pair distances for positive samples and maximize them for negative samples.

*2.5.1 Sample pairs construction.* We construct sample pairs based on local behavior feature $f_B$ and global interaction feature $f_I$. If $f_B$ and $f_I$ come from the same channel, the sample pair is positive, otherwise, the sample pair is negative. A sample pair $P$ is represented as:

$$P = (f_{B_{e_1}}, f_{I_{e_2}}, y), y = \begin{cases} 0, e_1 = e_2 \\ 1, e_1 \neq e_2 \end{cases} \tag{8}$$

where $y$ is the label indicating whether the sample pair is positive or negative, with 0 for positive and 1 for negative.

**Mitigation of false negative samples.** During the construction of sample pairs, it is possible to encounter channels from different host pairs that exhibit similar behavior, thereby resulting in same or very similar global or local features. When negative sample pairs are formed based on these two channels, false negatives are generated. Such false negative samples will negatively affect the training process and cause poor performance[36].

We propose a clustering-based strategy to mitigate this issue. To be specific, we first use the K-means algorithm to divide the local behavior features of the channels into $N_{cl}$ clusters and then construct negative sample pairs only among channels belonging to different clusters.

*2.5.2 Loss function.* To train the model, we first calculate the distance $y_d$ between $f_B$ and $f_I$ using a bilinear function. Then, we scale this distance to a range between 0 and 1 using a sigmoid function $\sigma$, and employ a cross-entropy loss function $F_{BCE}$ to calculate the training loss $\mathcal{L}$ based on the sample pair labels. The process is formulated as:

$$y_d = f_B W^d f_I \tag{9}$$

$$\mathcal{L} = F_{BCE}(y, \sigma(y_s)) \tag{10}$$

where $W^d$ is a learnable parameter.

## 2.6 Anomaly Detection

After training, the distance between local behavior feature and global interaction feature is small for a positive sample pair but big

for a negative pair. The anomalies in malicious channels manifest either in local or global features, or result from inconsistencies between these two types of features due to the malicious intent. Our well-trained ContraMTD model can map these anomalous features or intents onto a feature space distinct from that of previous normal behaviors, resulting in a shift in the distance distribution between the two types of features within either the positive or negative sample pairs, compared to the normal situation.

We determine whether a channel is malicious by holistically considering the distance between the two types of features in both the positive and negative sample pairs associated with that channel. If either the positive or negative sample pairs show an anomaly, the channel is labeled as malicious. By adopting this approach, we not only account for anomalies relative to past normal behavior in vertical perspective but also consider anomalies in comparison to other concurrent channels in horizontal perspective. The decision process is formalized as:

$$r = \psi(y_{d^p}) | \psi(\frac{1}{N_r} \sum_{i=1}^{N_r} y_{d^n}^{(i)}) \tag{11}$$

$$\psi(s) = \begin{cases} 0, & \mu_{y_d} - \gamma\sigma_{y_d} < y_d < \mu_{y_d} + \gamma\sigma_{y_d} \\ 1, & otherwise \end{cases} \tag{12}$$

where | denotes the 'or' operation, $y_{d^p}$ and $y_{d^n}$ are the distance of two features of positive and negative sample pairs respectively. $\psi(\cdot)$ is the function that determines whether a sample pair is anomalous. $\mu_{y_d}$ and $\sigma_{y_d}$ are the mean and the standard deviation of positive or negative sample pairs used to train the contrastive model. $\gamma$ is the threshold that can be adjusted manually.

The selection of negative sample pairs is random and can affect the final outcome. Therefore, we design a multi-round scoring approach that constructs multiple negative sample pairs for a channel and then averages the distances. $N_r$ is the number of rounds.

## 3 EXPERIMENTS

### 3.1 Setup

*3.1.1 Dataset.* To comprehensively evaluate the performance of the model, we conduct experiments on three datasets with different scales and application scenarios, including a self-collected dataset.

**Dataset description.** (1) *CICIDS*2018 [27]. The CICIDS2018 dataset contains network traffic from a simulated enterprise environment, featuring attacks like botnet and DoS, with around 500 machines involved. (2) *Real-MTU*. The Real-MTU dataset is self-made and focuses on encrypted malware traffic. It includes benign traffic from enterprise switches and malicious traffic from the Malware Capture Facility Project[2]. The malware types are adware, botware, miner, ransomware, and spyware. We replay this traffic to create the dataset. (3) *CICIoT*2023 [26]. The CICIoT2023 dataset features realistic IoT traffic with attacks like spoofing, and Mirai, collected from a topology composed of over 100 real IoT devices.

**Ethical concerns.** We obtain permissions from both the enterprise and individual users involved in the Real-MTU dataset collection process. Moreover, our work is limited to using general attributes such as packet length and does not involve parsing the packets in a way that would compromise user privacy.

*3.1.2 Metrics.* Our ContraMTD classifies network traffic into normal and malicious. Due to the inherent class imbalance between malicious and normal traffic, we use Precision (PR), Recall (RC), F1, and AUC as evaluation metrics in order to comprehensively and accurately assess the model's performance.

*3.1.3 Baseline.* To measure the improvements of our method, we use 7 state-of-art methods in 3 classes as baselines:

**Unsupervised malicious traffic detection method (US-MTD).** (1) *CIC-GMM* leverages the Gaussian Mixture Model for anomaly detection, taking CIC features[27] as its input. (2) Rosetta[34] trains a feature extractor based on packet length sequences. *Rosetta-IF* uses isolation forest to detect anomalies on these extracted features. (3) *Kitsune* [25] is an ensemble of autoencoders and it learns the pattern of a normal packet from multiple perspectives. (4) *CPS-Guard* [5] is an outlier-aware autoencoder. It calculates reconstruction losses and sets adaptive thresholds for outlier detection.

**Supervised malicious traffic detection method (S-MTD).** Automated Machine Learning [15] automates tasks like model and feature selection, and its accuracy can reach nearly 100% in supervised tasks. We implement *AutoML-MD* based on the AutoGluon [1] library to classify normal and malicious traffic using CIC features. For training, we use traffic that is either different from the malicious traffic in the test set or partially belongs to the same category.

**Attributed graph anomaly detection method (AGAD).** (1) *Dominant* [8] uses Graph Convolutional Networks GCNs for node embeddings and autoencoders for anomaly detection, considering both node attributes and graph structure. (2) *ANEMONE* [18] uses two GNN-based contrastive networks to learn the patch and context-level agreement, and then detects anomalies by statistical estimation.

### 3.2 Comparison with Baselines

*3.2.1 Overall performance.* Table 1 summarises the comparison results of ContraMTD and baseline methods on the three datasets. It can be observed that ContraMTD achieves the best performance on all metrics on the CICIDS2018 dataset, with F1 and AUC reaching 94.82% and 96.48%, respectively. On the other two datasets, ContraMTD ranks as the best or second-best across various metrics. The results on the three datasets with varying scales and application scenarios comprehensively demonstrate the effectiveness of our ContraMTD.

In terms of unsupervised malicious traffic detection methods, Rosetta-IF shows the most stable and best performance. This can be attributed to its feature extractor having been trained on large-scale network traffic, allowing it to adapt to various environmental changes. Similar to Rosetta-IF, ContraMTD also utilizes packet length sequences as input and compares them with other samples during the detection process. However, due to ContraMTD's learning of global features based on the topology of host interactive graphs, it outperforms Rosetta-IF with an AUC improvement ranging from 3.74% to 13.12%. Kitsune and CPS-Guard use autoencoder-based detection, but our method's AUC is at least 9.79% higher. Kitsune focuses on individual packets and ignores their correlation. CPS-Guard uses just one autoencoder, making it less effective. These limitations lead to their lower performance.

**Table 1: Overall comparison results with baseline methods (%)**

| Dataset | | CICIDS2018 | | | | Real-MTU | | | | CICIoT2023 | | | |
|---|---|---|---|---|---|---|---|---|---|---|---|---|---|
| Method | | PR | RC | F1 | AUC | PR | RC | F1 | AUC | PR | RC | F1 | AUC |
| US-MTD | CIC-GMM | 53.38 | 47.81 | 50.44 | 63.05 | 82.81 | 30.72 | 44.82 | 64.68 | 37.54 | 64.72 | 47.51 | 64.71 |
| | Rosetta-IF | 80.42 | 86.82 | 83.49 | 83.36 | 69.18 | 87.66 | 77.33 | 81.27 | 89.90 | 74.88 | 81.71 | 84.45 |
| | Kitsune | 74.91 | 91.62 | 82.42 | 86.69 | 80.84 | 71.28 | 75.75 | 79.96 | 17.63 | 85.72 | 29.24 | 56.62 |
| | CPS-Guard | 63.98 | 61.23 | 62.57 | 69.41 | 71.15 | 84.84 | 77.39 | 82.83 | 59.18 | 62.32 | 60.71 | 63.47 |
| S-MTD | AutoML-MD | / | / | / | / | **87.65** | 71.45 | 78.73 | 86.68 | **96.65** | 83.16 | **89.41** | **91.35** |
| AGAD | Dominant | 63.88 | 71.73 | 67.57 | 67.42 | 34.21 | **95.91** | 50.43 | 57.38 | 35.42 | 36.66 | 36.03 | 53.93 |
| | ANEMONE | 67.85 | 72.48 | 70.08 | 68.31 | 81.20 | 51.18 | 62.78 | 61.16 | 88.83 | 33.19 | 48.32 | 66.01 |
| Ours | ContraMTD | **96.09** | **93.56** | **94.82** | **96.48** | 70.02 | 95.69 | **80.64** | **94.14** | 83.69 | **86.35** | 84.99 | 88.19 |

\* **Bold** denotes the best results, and underline denotes the second-best results.
\* "/" indicates that the AUC is 50%, means that the result is a random guess.

For the supervised method AutoML-MD, it can be observed that it performs well in supervised scenarios. However, it fails on the CICIDS2018 dataset by labeling all samples as normal, essentially making random guesses. It performs better on other two datasets because it was trained on similar attack types. This suggests that supervised methods are sensitive to the variability between train data and test data, and are unstable when dealing with samples not seen in train data.

Compared to the two attributed graph anomaly detection methods, the performance of ContraMTD has shown a significant improvement. The AUC is 33.36% higher than Dominant and 27.77% higher than ANEMONE. Dominant and ANEMONE mainly target anomalies in citation networks or social networks, where the information spans a long time and the attributes and structures of nodes or edges are relatively stable. In contrast, host interactive graphs have more variable user behavior, more participants, and complex features. This makes detection harder and reduces Dominant's effectiveness. Both ANEMONE and our ContraMTD use contrastive learning, but ANEMONE ignores the relativity of anomalies at the time of detection and lacks horizontal contrast. ContraMTD improves on this by using horizontal and vertical comparisons, adapting to diverse and changing behaviors. Additionally, DE-GAT enhances ContraMTD's learning from host interaction graphs, boosting performance.

Moreover, we observe that the performance of nearly all methods on the CICIoT2023 dataset is the poorest among these datasets. This is due to that IoT traffic is different from standard enterprise network traffic. In IoT environments, there are a variety of non-standard communication protocols, coupled with diverse application scenarios and long activity cycles, making attack detection harder. Nonetheless, ContraMTD performs well by using specialized local and global features and a contrastive learning approach.

*3.2.2 Detailed performance.* To better understand the performance of ContraMTD, we further look into the AUC of ContraMTD and baselines for each category of malicious traffic on the three datasets, and the heatmaps are shown in Figure 2.

As shown in Figure 2(a), ContraMTD can detect most types of malicious traffic in the CICIDS2018 dataset, except for Infiltration.

One reason behind the poor AUC may be attributed to that infiltration attacks are quite stealthy and occur occasionally. In addition, as illustrated in Figure 2(b), ContraMTD is capable of identifying the malicious traffic generated by each type of malware. The lowest performance is observed in detecting ransomware, where the AUC is 91.86%. As shown in Figure 3, our model effectively separates normal and malicious channels based on the distances of their global and local features within both positive and negative sample pairs, thereby achieving excellent detection performance.

Besides, it can be observed that ContraMTD has a high AUC on flood-type traffic, like BruteForce, DoS in the CICIDS2018 dataset, as well as Spoofing in the CICIoT2023 dataset. This is because we use channels as detection objects, and the aggregation nature of flood-type attacks amplifies their anomalous behaviors. The reason for the relatively low AUC of ContraMTD in detecting DDoS in CICIoT2023 is primarily due to the main type being Slowloris[26], which occupies resources by slowly sending data or intentionally leaving transmissions incomplete and need to monitor the state of connection for effective detection.

## 3.3 Complexity Analysis

To fully evaluate the trade-off between performance and complexity, we present the number of model parameters, the number of floating point operations (FLOPs), and time overhead in Table 2. For fair comparisons, we standardize the evaluation metric to the time required to process each session when assessing time overhead. Besides, the batch size is 1 when calculating FLOPs, and the FLOPs are averaged over each node or edge for graph based methods.

We can find that the time overhead of ContraMTD is lower than that of Kitsune and Rosetta-IF, two models with marginally better performance, and slightly higher than that of CPS-Guard. In general, the computational complexity and time consumption of ContraMTD are slightly higher, but it achieves considerable performance improvements with only a modest increase in complexity.

## 3.4 Ablation Experiment

To verify the contributions of each component in ContraMTD, we conduct ablation studies on the CICIDS2018 dataset, and the results are shown in Table 3.

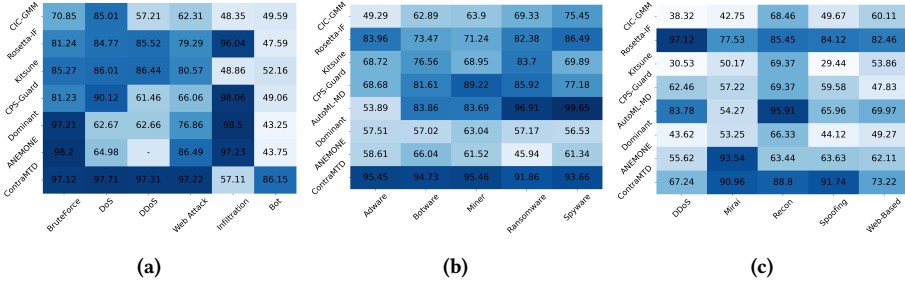

Figure 2: The heatmaps of AUC for each category of malicious traffic on (a) CICIDS2018, (b) Real-MTU and (c) CICIoT2023.

Figure 3: Distribution of the distance between local and global features of sample pairs on Real-MTU.

Table 2: Model parameters, FLOPs and time overhead

| Method | Parameters | FLOPs | Times (ms) |
|---|---|---|---|
| CIC-GMM | / | / | 1.33e-02 |
| Rosetta-IF | 1.12e+07 | 1.42e+07 | 8.53e-01 |
| Kitsune | / | / | 7.19e+00 |
| CPS-Guard | 6.06e+03 | 1.09e+04 | 1.74e-01 |
| AutoML-MD | / | / | 1.13e-02 |
| Dominant | 8.14e+03 | / | 9.67e-02 |
| ANEMONE | 1.81e+04 | 3.14e+04 | 1.38e-01 |
| ContraMTD | 1.12e+05 | 1.22e+05 | 4.76e-01 |

* "/" indicates that the number is either non-existent or inaccessible.

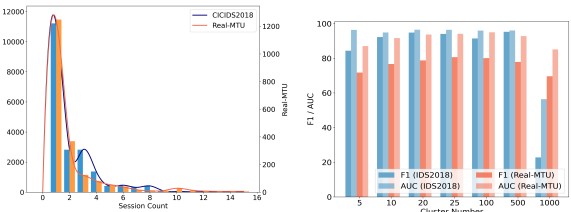

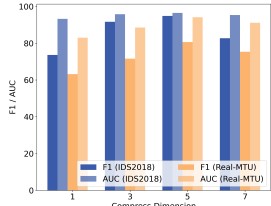

Figure 4: The distribution of the number of sessions in a channel.

Figure 5: F1 and AUC with different cluster number.

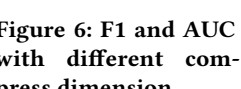

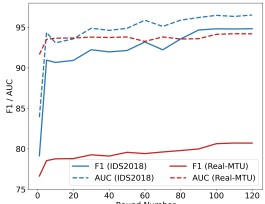

Figure 6: F1 and AUC with different compress dimension.

Figure 7: F1 and AUC with different round number.

We can find that the utilization of SEC significantly enhance the precision, subsequently leading to great improvement in F1 score. Besides, it is worth noting that when CNN is removed, the performance declines significantly. This can be attributed to the role of CNNs in high-dimensional feature extraction and scaling, functionalities that the subsequent portions of the model are unable to fulfill. Replacing CNN with LSTM did not restore the original performance, mainly because the low number of sessions in each channel limits the advantages that LSTMs usually offer. Moreover, the use of DE-GAT also notably improves both F1 score and AUC. Substituting DE-GAT with GraphSAGE[13] and GCN[20] leads to reduced performance, largely due to DE-GAT's attention mechanism effectively focusing on pivotal sessions and channels.

Additionally, the usage of the cluster strategy results in a 1.02% improvement in AUC for ContraMTD, implying that our approach can effectively mitigate the issue of false negatives samples in contrastive learning.

In detection stage, negative samples are more influential, but positive sample pairs also make substantial contributions. The AUC still reaches 91.84% when only positive sample pairs are involved in decision.

## 3.5 Sensitivity Analysis

To investigate the impact of hyper-parameters on performance, as well as the relationship between network scale and performance, we conduct experiments on the CICIDS2018 dataset and the Real-MTU dataset. These datasets represent enterprise networks at different scales, with Figure 4 illustrating the distribution of session counts per channel.

**Cluster number.** Figure 5 shows that the number of clusters $N_{cl}$ in constructing sample pairs doesn't greatly affect performance. This is because clustering mainly separates samples from different classes. However, too many clusters can lead to many clusters with only one sample, which may reduce performance by causing repetition in negative sample pairs for clusters with many samples.

**Compress dimension.** Figure 6 illustrates that as the compression dimension $N_{cp}$ goes from 1 to 5, both F1 and AUC improve. However, performance declines when dimensions increase beyond that. This is because while more dimensions capture more information, they also add noise and slow down detection. Most channels have 5 or fewer sessions, making up over 90% of the cases.

**Round number.** Figure 7 shows that F1 and AUC improve quickly as the number of scoring rounds $N_r$ goes from 1 to 5. After

**Table 3: Ablation Study of Key Components in ContraMTD on CICIDS2018 (%).**

| Method | SEC | CNN | DE-GAT | cluster | P | N | PR | RC | F1 | AUC |
|---|---|---|---|---|---|---|---|---|---|---|
| w/o SEC | × | ✓ | ✓ | ✓ | ✓ | ✓ | 53.73 $\downarrow_{42.36}$ | 99.98 $\uparrow_{6.42}$ | 69.90 $\downarrow_{24.92}$ | 93.15 $\downarrow_{3.33}$ |
| w/o CNN | ✓ | × | ✓ | ✓ | ✓ | ✓ | 72.93 $\downarrow_{23.12}$ | 49.67 $\downarrow_{43.89}$ | 59.09 $\downarrow_{35.73}$ | 73.37 $\downarrow_{23.11}$ |
| w/ LSTM | ✓ | ○ | ✓ | ✓ | ✓ | ✓ | 66.20 $\downarrow_{29.89}$ | 97.94 $\uparrow_{4.38}$ | 79.00 $\downarrow_{15.82}$ | 94.99 $\downarrow_{1.49}$ |
| w/o DE-GAT | ✓ | ✓ | × | ✓ | ✓ | ✓ | 79.15 $\downarrow_{16.94}$ | 92.24 $\downarrow_{1.32}$ | 85.19 $\downarrow_{9.63}$ | 92.94 $\downarrow_{3.54}$ |
| w/ GCN | ✓ | ✓ | ○ | ✓ | ✓ | ✓ | 75.65 $\downarrow_{20.44}$ | 97.31 $\uparrow_{3.75}$ | 85.13 $\downarrow_{9.69}$ | 96.17 $\downarrow_{0.31}$ |
| w/ GraphSAGE | ✓ | ✓ | ○ | ✓ | ✓ | ✓ | 68.01 $\downarrow_{28.08}$ | 99.96 $\uparrow_{6.40}$ | 80.95 $\downarrow_{13.87}$ | 96.24 $\downarrow_{0.24}$ |
| w/o cluster | ✓ | ✓ | ✓ | × | ✓ | ✓ | 82.46 $\downarrow_{13.63}$ | 92.81 $\downarrow_{0.75}$ | 87.33 $\downarrow_{7.49}$ | 94.84 $\downarrow_{1.64}$ |
| w/o P | ✓ | ✓ | ✓ | ✓ | × | ✓ | 92.78 $\downarrow_{3.31}$ | 90.58 $\downarrow_{2.98}$ | 91.67 $\downarrow_{3.15}$ | 94.73 $\downarrow_{1.75}$ |
| w/o N | ✓ | ✓ | ✓ | ✓ | ✓ | × | 78.46 $\downarrow_{17.63}$ | 87.51 $\downarrow_{6.05}$ | 82.74 $\downarrow_{12.08}$ | 91.84 $\downarrow_{4.64}$ |
| ContraMTD (default) | ✓ | ✓ | ✓ | ✓ | ✓ | ✓ | 96.09 | 93.56 | 94.82 | 96.48 |

\* "N" and "P" respectively refer to the participation of negative and positive sample pairs in the anomaly detection process.

\* "○" indicates that the corresponding component in ContraMTD is replaced.

that, the growth slows and stabilizes when more than 100 negative samples are used. Due to the randomness in the selection of negative samples, fewer samples are more error-prone, but as more are added, the model becomes more fault-tolerant and performance nears its peak.

## 4 DISCUSSION

In this section, we discuss some potential limitations and challenges of our ContraMTD.

**Anti-Evasibility.** Attackers could potentially evade our security method by mimicking normal network traffic. Achieving this would require complex tasks such as extensive data collection to understand the learned contrastive model and manipulating traffic from multiple hosts, which are difficult to execute. However, if attackers are already inside the system before we deploy our method, detection becomes infeasible.

**Scalability.** We conduct experiments across enterprise networks of varying scales and demonstrate the algorithm's feasibility. However, scaling to networks with tens of thousands of devices poses challenges in computation, storage, and optimization due to the graph models used. Future work could address these by breaking down large graphs into smaller ones for separate learning, and possibly using federated learning methods.

## 5 RELATED WORK

**Malicious network traffic detection.** In supervised approaches, the primary workflow involves extracting statistical features or utilizing deep learning models to capture the latent feature of traffic, and then train a classifier for categorization. Jordan et al. [16] leverages AutoML for fully automated traffic analysis. Han et al. [14] employs n-gram and Transformer to extract structural and temporal of traffic. Besides, Wang et al. [33] combine contrastive learning and federated learning to boost detection capability in a supervised way.

Unsupervised approaches can be divided into vertical comparison based and horizontal comparison based. In vertical comparison, one approach is to create a baseline model using statistical features to identify abnormal data. Wang et al. [32] analyzes the byte distribution of payloads on specific ports. Another approach uses autoencoders to reconstruct normal traffic, and large reconstruction errors are considered malicious. Mirsky et al. [25] and Catillo et al. [5] develop autoencoders targeting packets and flows, respectively. Horizontal comparison based methods process multiple samples at the same time, and identify anomalous data that significantly deviate from the majority. Zhang et al. [35] propose a on-demand evolving isolation forest to detect malicious traffic.

**Graph anomaly detection.** Anomaly detection on attributed graphs aims to identify anomalous nodes from graph data. Sambaran et al. [3] employs an autoencoder to detect attribute anomalies, and Ding et al. [8] also utilizes an autoencoder, simultaneously reconstructing the graph's attribute and structural information. Contrastive learning has also been frequently adopted in this domain. Liu et al. [24] detects anomalies by calculating the relationship between nodes and their neighborhoods, and Jin et al. [18] leverages patch and context-level agreement of nodes. Duan et al. [10] proposes a multi-view multi-scale contrastive learning framework, introducing subgraph-subgraph contrast.

## 6 CONCLUSION

In this paper, we apply contrastive learning to unsupervised malicious traffic detection for the first time, and propose a novel method called ContraMTD. ContraMTD learns the local behavior feature and the global interaction feature of channels, and employs contrastive learning to learn normal patterns by leveraging the purpose consistency between them, then detects anomalies through multiround scoring. By doing these, ContraMTD can detect malicious traffic from both horizontal and vertical perspectives. We evaluate the performance of ContraMTD on three datasets, demonstrating that ContraMTD can effectively detect malicious traffic without label. In the future, we will further investigate the anti-evasibility and scalability of ContraMTD and work to improve its efficiency.

## ACKNOWLEDGMENTS

We thank the anonymous reviewers for their constructive feedback and valuable insights that enhanced our paper.

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
