# OpenReview forum: "ContraMTD: An Unsupervised Malicious Network Traffic Detection Method based on Contrastive Learning"
_ACM.org/TheWebConf/2024/Conference — TheWebConf24_

### Official Review · Reviewer_rmxa · 2023-11-19

**Novelty:** 5
**Technical Quality:** 5

**Review:**

# Summary

The paper introduces a novel anomaly detector designed for intrusion detection. It achieves this by extracting both local and global features, which are then utilized in contrastive learning to model normal behavior. This approach enables the detection of unusual network traffic. The method is benchmarked against seven state-of-the-art approaches using three distinct datasets. It demonstrates comparable performance to existing methods and even surpasses them on the CICIDS2018 dataset.

# Strengths

+ Novel approach based on contrastive learning
+ Evaluation on three different datasets
+ Comparison with 7 state-of-the-art methods

# Weaknesses

- Only slight improvement compared to state of the art
- No comparison with simple baseline methods
- Code not publicly available

# Detailed comments

The reliable detection of malicious traffic using machine learning remains a significant and unresolved challenge in the field of computer security research. The proposed approach, which involves combining local and global features and employing contrastive learning to develop an anomaly detector, is a concept that I find compelling. Specifically, the use of Graph Attention Networks (GATs) to model global interactions is a promising direction that merits further exploration.

The paper conducts a comprehensive evaluation, including comparisons with several state-of-the-art methods across three distinct datasets. While it only significantly outperforms other approaches on one of these datasets (CICIDS2018), the authors demonstrate that their method is less complex than related approaches that show a similar performance. Additionally, they provide an analysis of the effectiveness and impact of the individual components of their system.

Besides providing only moderate performance improvements, the paper also exhibits other shortcomings:

`Simple baselines`. While the authors compare their approach with several other methods, the paper lacks a comparison with a simple approach, such as [1], to show that complexity introduced by DL-based approaches is required to detect the attacks in the used datasets (see [2]). I think the paper would benefit from such a comparison, which would increase the expressiveness of the obtained results.

`Missing information`. The experimental setting lacks some essential information that allow assessing the soundness of the conducted experiments. In particular, they do not explicitly describe how they pre-processed and split the used datasets to avoid common biases, such as spatial and temporal bias (see [2] and [3]).

`Adversarial examples`. The paper does not provide any experiment with an adaptive attacker. Thus, it remains unclear whether the proposed system can be easily circumvented or not. However, the authors at least discuss this issue in Section 4.

`Reproducability`. It is unclear whether the code will be made publicly available. This would allow other researchers to verify the results and conduct more research in this direction.


Despite these weaknesses, I think the paper combines several interesting ideas for detecting anomalous/malicious traffic and examines their effectiveness in a thorough evaluation. Although the results are not groundbreaking, I still think that it extends the state of the art.

### Additional comments

- Reference 25 has been published at a peer-reviewed conference (NDSS). The authors should replace the reference with the conference version.

### References

1. Wang et al., Anagram, RAID 2006
2. Pendlebury et al., Tesseract, USENIX Security 2019
3. Arp et al., Dos and Don'ts of Machine Learning in Computer Security, USENIX Security 2022

**Questions:**

- In Section 2.3, the authors state that they reshape the local features into a 2-dimensional matrix and then feed into a CNN. From my understanding, the reshaping-step does not make much sense as it introduces arbitrary (horizontal) dependencies between features, depending on the selected dimensionality of the matrix. I recommend the authors to use a 1-D CNN instead and/or explain better why using a 2D-CNN makes sense in this case.
- In Secton 3.1, they mention that the Real-MTU dataset has been created by merging benign and malicious traffic. This might have led to shortcut artifacts that might over-estimate the system's detection capabilities. Did the authors take any steps to ensure that this is not the case? If yes, what actions did they perform?

**Ethics Review Description:**

-

**Reviewer Confidence:**

3: The reviewer is confident but not certain that the evaluation is correct

**Scope:**

3: The work is somewhat relevant to the Web and to the track, and is of narrow interest to a sub-community

---

### Official Review · Reviewer_Ppbr · 2023-11-22

**Novelty:** 3
**Technical Quality:** 3

**Review:**

This paper proposes a new method called ContraMTD, which applies contrastive learning to detect malicious traffic in an unsupervised manner. ContraMTD includes five modules responsible for the following tasks: dividing traffic in channels representing interactions among different hosts, extracting features for local hosts, extracting interaction features across different hosts, using contrastive learning to construct positive and negative sample pairs for model training, and a multi-round scoring scheme for anomaly detection. K-means clustering is used to form positive and negative sample pairs in model training. For performance evaluation, three datasets are used. Seven existing techniques falling into three categories are used for performance comparison. The paper includes the results from performance comparison, complexity analysis, ablation study, and sensitivity analysis.

Strengths:
+ The work has applied contrastive learning for malware traffic detection for which the idea is less explored previously.
+ The proposed technique has combined both local behavior features and global interaction features for anomaly detection.
+ The evaluation work used three datasets and seven existing techniques for performance comparison.

Weaknesses:
- For the three datasets used, the proposed technique has high false alarm rates on two of them (indicated by the precisions).
- The proposed method uses the K-means clustering results to guide the formation of positive and negative sample pairs for contrastive learning. Although this allows unsupervised learning, it also makes the performance of contrastive learning sensitive to the K-means method used.

**Questions:**

* In Table 1, it shows that for both Real-MTUa nd CICIoT2023 datasets, the precision scores are poor for the proposed method. This indicates that it has a high false alarm rate, which makes it difficult to be deployed in practical environments as an anomaly-based intrusion detection system.

* Is the scheme proposed meant for online or offline intrusion detection? If it's for online intrusion detection, it might be helpful to present the computational overhead in constructing the global interaction graph. If it's for offline intrusion detection, it might be helpful to show the storage needed to store traffic traces. The operational mode of the proposed scheme is unclear from the description in the paper.

**Reviewer Confidence:**

2: The reviewer is willing to defend the evaluation, but it is likely that the reviewer did not understand parts of the paper

**Scope:**

2: The connection to the Web is incidental, e.g., use of Web data or API

---

### Official Review · Reviewer_ws9d · 2023-11-23

**Novelty:** 5
**Technical Quality:** 4

**Review:**

This paper proposes a novel method ContraMTD based on contrastive learning, which comprehensively considers both vertical and horizontal perspectives. ContraMTD extracts local behavior features and global interaction features from normal network traffic by proposed SEC and DE-GAT respectively, then employs contrastive learning to learn the relationship, especially consistency between them, and finally detects malicious traffic through a multi-round scoring approach.
Some major concerns,
1. The experimental results are questionable. According to Table 1, S-MTD is the most competitive baseline, and its results have been reported on both the Real MTU and CICIoT2023 datasets. Why is it a random guess on the CICIDS2018 dataset?

2. For the proposed Graph Double Edge Attention Network for global feature extraction, it seems that similar models already exist. Authors are encouraged to provide relevant supporting literature or highlight new design features.

**Questions:**

1. The experimental results are questionable. According to Table 1, S-MTD is the most competitive baseline, and its results have been reported on both the Real MTU and CICIoT2023 datasets. Why is it a random guess on the CICIDS2018 dataset?

2. For the proposed Graph Double Edge Attention Network for global feature extraction, it seems that similar models already exist. Authors are encouraged to provide supporting literatures or highlight new design features.

**Reviewer Confidence:**

4: The reviewer is certain that the evaluation is correct and very familiar with the relevant literature

**Scope:**

4: The work is relevant to the Web and to the track, and is of broad interest to the community

---

### Official Review · Reviewer_yr8i · 2023-11-23

**Novelty:** 4
**Technical Quality:** 5

**Review:**

**Summary:**

The paper proposes a novel method ContraMTD based on contrastive learning, which comprehensively considers both vertical and horizontal perspectives. ContraMTD extracts local behavior features and global interaction features from normal network traffic by proposed SEC and DE-GAT respectively, then employs contrastive learning to learn the relationship, especially consistency between them, and finally detects malicious traffic through a multi-round scoring approach.

**Pros:**

The article is written in a clear and accessible manner, making it easy for readers to understand.
The paper conducts a substantial number of experiments, and based on the experimental results, the proposed model appears to outperform some existing methods.

**Cons:**

1.In fact, contrastive learning, including supervised contrastive learning, has been widely applied to specific domains in recent years, such as software defect detection. However, introducing it into other fields may lack a certain degree of innovation. This suggests that utilizing contrastive learning in new domains may require deeper consideration and innovative approaches to ensure its effectiveness and adaptability.
2.Experiments on more public datasets should be supplemented.

**Questions:**

I hope to ask the authors' responses to my questions outlined in the weaknesses section.

**Reviewer Confidence:**

3: The reviewer is confident but not certain that the evaluation is correct

**Scope:**

3: The work is somewhat relevant to the Web and to the track, and is of narrow interest to a sub-community

---

### Official Review · Reviewer_utX6 · 2023-11-25

**Novelty:** 3
**Technical Quality:** 3

**Review:**

This paper proposes an unsupervised model to detect malicious network traffic. The main novelty lies in the analysis of network channels from both a vertical and horizontal perspective, and the use of contrastive learning to model normal patterns.

Pros:
- The presentation quality is sufficient.
- To the best of my knowledge, the proposed methodology is novel.

Cons:
- The related work section is insufficient. (too brief and not comprehensive)
- The paper lacks technical details, making it difficult to assess the soundness of the proposed methodology. As an example, many choices are not well motivated, such as considering channels (rather than sessions or flows).
- The adversary model is missing.
- Many sentences are unclear, such as “we reply this traffic to create the dataset”.
- Apparently, the proposed model uses packet size as one of the main features. This is good when the adversary collects network traffic in the target machine, as network packets are reassembled for TCP flows and the packet dimensions vary considerably. However, if the data is collected outside the target machine (as in a standard and reasonable adversary model), this feature might lose importance.
- The authors claimed that one of the used datasets is “self-collected”. However, it is unclear which one of the three datasets is self-collected, with all the information related to the data collection and the description of the dataset completely missing.
- The proposed model outperforms baseline techniques on old data only (data from 5 years ago, possibly no longer representative of today’s network traffic). The authors should address this point, investigating the reason behind it.

**Questions:**

- What are the advantages of aggregating network packets in channels, rather than sessions or flows? (a detailed motivation is needed, possibly with example, rather than general a sentence like “a channel contains more comprehensive behavior than flow”)
- What are the adversary capabilities? Where is the adversary located? (same host, same LAN, etc.)
- Which one of the three datasets is self-collected? How did you collect the data?

**Reviewer Confidence:**

3: The reviewer is confident but not certain that the evaluation is correct

**Scope:**

4: The work is relevant to the Web and to the track, and is of broad interest to the community

---

### Decision · Program_Chairs · 2024-01-22

**Decision:**

Accept

**Comment:**

The paper scores relatively well as per novelty and technical quality, especially when these dimensions are assessed with respect to the reviewers' self-assessed confidence. Though there are some relevant elements that need to be addressed (e.g. the adversary model is missing), the interactions between the authors and the reviewers led to a more positive appreciation of the contribution.

 ---